# Diagnostic Yield of Neuroimaging for Headache in a Pediatric Emergency Department: A Single Tertiary Centre Experience

**DOI:** 10.3390/healthcare11060913

**Published:** 2023-03-22

**Authors:** Mohammed Almuqbil, Saud Abdulaziz Alsayed, Amer Mohammed Almutairi, Khalid Mohammed Aladhadh, Abdullah Omar Alghannami, Mohammed Almutairi

**Affiliations:** 1College of Medicine, King Saud Bin Abdulaziz University for Health Sciences (KSAU-HS), Riyadh 11481, Saudi Arabia; 2Pediatric Emergency Department, King Abdullah Specialist Children Hospital (KASCH), National Guard Health Affairs (NGHA), Riyadh 52569, Saudi Arabia; 3King Abdullah International Medical Research Center (KAIMRC), Ministry of National Guard, Riyadh 11481, Saudi Arabia

**Keywords:** CT scan, emergency department, headache, MRI, pediatric

## Abstract

Objectives: This study aimed to examine headache neuroimaging findings among the pediatric population visiting the emergency department in Saudi Arabia. Methods: This was a cross-sectional retrospective study of pediatric patients who presented to the emergency department with a headache as their primary complaint. Data were extracted from the electronic medical files of the patients at King Abdullah Specialized Children Hospital (KASCH) between 2015 and 2020. The diagnosis of headache was confirmed using a computerized tomography (CT) scan or magnetic resonance imaging (MRI) upon the patients’ presentation. Results: A total of 263 patients met the inclusion criteria, and their data were extracted. The CT scans were abnormal in 50% of the patients. The MRI showed abnormal findings for 26% of the patients. CT scans and MRI identified that abnormalities were predominantly among patients with the secondary type of headache. The most common abnormal findings on CT were sinusitis (16%), masses (7%), and hydrocephalus (7%). The most common abnormal findings on MRI were masses (8%), cysts (5%), and hydrocephalus (3%). Of all patients with headaches, 10% had a prior diagnosis of headache, and 12% had a family history of headache. A significantly higher percentage of patients with secondary headache were prescribed NSAID and required admission compared to patients with primary headache (*p* ≤ 0.05). There was no statistically significant differences in the proportion of patients diagnosed with primary and secondary headache in terms of their neurological examination and headache types (*p* = 0.43). Conclusions: Neuroimaging is essential for diagnosing headaches in children. Headaches were associated with sinusitis in children. The secondary type was more likely to have abnormal CT and MRI results. Primary type headaches were more common in those with a family history. CT scans and MRIs are needed when a headache is accompanied by an abnormal clinical evaluation. Neuroimaging and mild CT usage may be explored if there are clinical abnormalities or family history.

## 1. Introduction

A headache is a common presentation that affects people of all ages and accounts for 0.5 to 4.5 percent of all visits to the emergency room [1]. The occurrence of headaches increases with age, and the causes vary from primary to secondary headache with the highest prevalence in females aged 16 to 18 [2]. Nevertheless, headaches are a common ailment in children and one of the most frequent reasons for visits to the Pediatric Emergency Department (ED) [3]. The reported prevalence of headaches among school-aged children varies widely, from 6% to 82% depending on age and the defining criteria, and steadily increases towards adolescence [4,5]. Previous estimates indicate that up to 75% of children would experience at least one episode of major headache before the age of 15, and that up to 28% of teenagers will describe a migraine [6,7]. Approximately 1% of pediatric ED visits are attributed to headaches, according to some experts [8,9,10,11]. The vast majority of headache episodes in children are benign and self-limiting, with a modest prevalence of subsequent forms due to significant organic reasons (4–20%) [2,8,9,11,12,13,14]. In Saudi Arabia, around 60% of school-aged children have experienced at least one episode of headache [15]. In a retrospective study conducted in Italy, 184 patients (8.8%) with a headache complaint younger than 18 years old underwent neuroimaging, and 7.1% of them revealed a pathological abnormality [3]. In a 2013 systematic analysis conducted in Greece, 14.6% of 3026 pediatric patients who underwent computerized tomography (CT) and magnetic resonance imaging (MRI) scans showed aberrant findings. Chiari I malformation, sinusitis, and arachnoid cysts were the most prevalent anomalies [16]. In Iran, a 2018 cross-sectional study of pediatric patients younger than 12 years of age revealed that 24.3% of 217 children who had CT scans had abnormal findings, with hematomas and masses being the most common [17].

Primary headaches are not directly caused by structural defects, toxic exposure, or infections; the most common types of headaches in children are migraine (with and without aura) and tension-type headache. Secondary headaches are directly attributable to another clinical condition (such as infections, vascular or nonvascular intracranial illnesses, or psychiatric disorders) and resolve after treatment of the underlying cause [3]. The occurrence of secondary headaches owing to serious or life-threatening reasons is minimal, but their detection is a significant goal of the Pediatric ED, along with the provision of rapid, safe, and efficient pain relief. Consequently, the diagnostic approach to pediatric headache in the ED is of utmost importance. In 2013, the American Academy of Neurology (AAN) Practice Guidelines gave diagnostic approach recommendations for children and adolescents with recurrent headaches, with no particular comment about patients in the ED [5]. Moreover, despite the AAN guidelines and multiple studies indicating that neuroimaging is unnecessary to rule out significant intracranial abnormalities if no other neurologic symptoms are present, the CT scan is still commonly utilized for children with headache [2,13,16,18,19]. To obtain a proper diagnosis of headache and identify whether the headache is primary or secondary, a physician must take an accurate medical history, conduct a careful neurological examination, and obtain brain imaging studies, including CT scans and MRI as needed. CT imaging is one of the most common tools for evaluating headaches in children, helping to evaluate the face, skull, and sinuses by ruling out secondary causes of headaches, such as bleeding, brain tumors, hydrocephalus, etc. It can effectively indicate brain injuries and bleeding in emergency situations, saving patients from significant problems or death [20]. Any patient with progressive symptoms, bleeding tendency, severe headache, with a Glasgow Coma Score (GCS) < 13, and those on anticoagulants, or experiencing any features that may suggest meningoencephalitis, such as papilledema and neck stiffness, should undergo a CT scan [17].

To the best of our knowledge, minimal local research has been carried out that provided neuroimaging findings in children with headaches presented to the emergency department. The most recent local study conducted in Al Kharj city in Saudi Arabia in 2017 included 210 adult patients complaining of chronic headache or recurrent episodes of headache over one month. The authors of the research concluded that CT scans should not be used in the absence of additional clinical symptoms [21].

The main aim of this study is to assess neuroimaging findings in pediatric patients coming to the emergency department with a headache as a chief complaint. A further purpose of this study was to describe the physical examinations and management procedures for pediatric patients with primary and secondary headaches.

## 2. Methods

### 2.1. Study Design

From April 2015 to April 2020, a retrospective cross-sectional study was undertaken at the emergency department (ED) of King Abdullah Specialized Children’s Hospital (KASCH), Riyadh, Saudi Arabia. The BESTCare 2.0A System, which is a hospital information system, retrieved information from the patients’ electronic medical records (HIS).

### 2.2. Study Population

Inclusion criteria were any patient presenting to the ED with a headache as the primary complaint and an onset age between 3 and 15 years of age, excluding any patient with a headache that was preceded by trauma (i.e., traumatic brain injury).

### 2.3. Study Variables

The variables of interest that were collected in this study included basic demographic data (gender, age, and weight), a family history of headache, a history of chronic diseases, presenting symptoms, and physical and neurological examination at presentation. We also determined if the patient subsequently underwent an MRI for the same reason to determine if any lesions were missed by the CT scan. We analyzed each patient’s investigations and inpatient–outpatient medications to determine the best way to treat a child with an acute headache in the ED. The ordering of the CT and MRI was coordinated by the emergency department physician. The diagnosis and classification of headache types recorded in the ED were based on the third edition of the International Classification of Headache Disorders (ICHD-3), which was released by the International Headache Society (IHS) in 2013 [22]. 

### 2.4. Ethical Approval

The study protocol was approved by the King Abdullah International Medical Research Center Institutional Review Board (IRB) (RC20/175/R). Due to the retrospective nature of the data, patient consent was not required; nonetheless, no personally identifying data were obtained, allowing for the protection of patient privacy and confidentiality.

### 2.5. Data Analysis

The data were entered using Microsoft Excel and exported to SPSS (V23) for analysis. The quantitative variables (e.g., age) were presented as a confidence interval with mean and standard deviation (mean ± SD). Categorical data, such as gender and abnormal neuroimaging findings among patients, were presented as frequency and percentages (%). A Chi-squared test was used to assess the difference in the proportion of patients with primary and secondary outcomes in terms of their outcome variables (physical examination, radiological findings, and their management). A *p*-value of less than 0.05 and a confidence interval of 95% was declared to be statistically significant for all statistical tests.

## 3. Results

Among the 263 pediatric patients who presented to the ED with a headache complaint, 152 (58%) were males and 111 (42%) were female; the mean age was 11.6 (SD = 3.3) years; and the mean age of disease beginning was 9.3 (SD = 3.3) years. A total of 25 (10%) of the 263 patients had a prior diagnosis of headache, and 31 (12%) had a family history of headache; 109 (41%) had a history of chronic disorders, with neurological disorders being the most prevalent (21%); other disorders not included in the criteria accounted for 23% (Table 1), with Chiari malformation, adenoid hypertrophy, and cistern lipoma being the leading three causes. 

A total of 260 patients had complete data in terms of physical examinations, CT and MRI findings, and management. The type of headache was also investigated among the study patients: 202 (78%) were identified with secondary type, and 58 (22%) with primary type. The percentages of migraine types were migraine without aura (66%), unspecified (22%) migraine with aura (9%), trigeminal autonomic cephalgia (2%), and tension-type (2%) headaches. Patients with a positive family history of headache had a considerably greater incidence of primary headaches (*p* < 0.001), and patients with chronic illnesses had a significantly greater frequency of secondary headaches (*p* < 0.001); see Figure 1. Around 52% of the patients were admitted, primarily for headaches of secondary origin. In 11% of the 263 individuals, a chronic headache was present (Table 2). The most common co-existed abnormalities were motor, cerebellar, and gait.

Computed tomography scans and MRI identified abnormalities were predominantly found among patients with the secondary type of headache. The CT scan was normal in 129 (50%) patients and abnormal in 131 (50%) patients, with 14 (24%) incidental abnormal findings in patients with known cases of primary type headache and 117 (58%) secondary type headache. Sinusitis (n = 42, 16%), masses (n = 18, 7%), hydrocephalus (n = 17, 7%), ventricular dilatation (n = 10, 4%), edema (n = 6, 2%), infarction (n = 2, 1%), VP shunt dysfunction (n = 5, 2%), and calcification (n = 4, 2%) were the predominant radiological findings, with some patients having multiple abnormalities. Nine individuals with sinusitis had headaches of the primary type, followed by unclassified findings. A total of 26.0% of the MRI scans were abnormal. These abnormal findings primarily consisted of masses (n = 21, 8%), cysts (n = 12, 5%), hydrocephalus (n = 8, 3%), hematoma (n = 3, 1%), hypoplasia of the corpus callosum (n = 2, 1%), and sinusitis (n = 2, 1%), as well as other unclassified findings of the secondary type; only two (3%) patients exhibited unspecified MRI abnormalities of the primary type (Table 3). The most common co-existed abnormalities were mass and hydrocephalus.

Antiemetics, NSAIDs, and acetaminophen were among the medications administered in the ED. A total of 118 (45%) patients were administered acetaminophen, the majority of whom (80%) were classified as secondary type. A total of 76 (29%) patients were prescribed NSAIDs, the majority of whom were diagnosed with secondary type (68%). Antiemetics were administered to 39 (15%) patients, with the secondary type constituting the majority (74%) of those receiving treatment. Blood cultures were performed on 84 (32%) patients, the majority of whom were secondary type (87%) patients; urine cultures were performed on 84 (32%) patients. The most prevalent outpatient medications (*p* < 0.05) were antibiotics and nonsteroidal anti-inflammatory drugs. A significantly higher percentage of patients with secondary headache were prescribed NSAID and required admission compared to patients with primary headache (*p* ≤ 0.05), Table 4.

There was no statistically significant difference in the proportion of patients diagnosed with primary and secondary headache in terms of their neurological examination and headache kinds (*p* = 0.43). A total of 155 (60%) individuals had normal neurological examinations, while 100 (39%) had abnormal examinations. Motor abnormalities were among the most common aberrant findings in both the primary and secondary groups of patients with abnormal neurological tests, accounting for 24 (9%) patients, followed by mental status alterations in 14 (5%) patients (Table 2).

## 4. Discussion

The third most common reason for visits to pediatric emergency rooms is headache (ED). A comprehensive analysis found that benign diseases that often go away on their own or with the right pharmacological care are to blame for the majority of headaches in children assessed in emergency departments [23]. This study evaluated neuroimaging findings in pediatric patients who presented to the emergency department with headache as their primary complaint. The characterization of physical examination and management practices for pediatric patients with primary and secondary headaches was another goal of this study. In this study, the most common abnormal findings on CT were sinusitis, masses, and hydrocephalus. The most common abnormal findings on MRI were masses, cysts, and hydrocephalus. CT scan- and MRI-identified abnormalities were predominantly found among patients with secondary type of headache, and patients with secondary type headache required hospital admission in a higher proportion than those with primary type headache.

In our study, secondary headache was the most prevalent type of headache in children, and 41% of the patients had a history of chronic illness, primarily respiratory and neurological conditions. The anomalies found during a neurological assessment included motor, gait, and sensory abnormalities, as well as altered mental status and language alterations. Around 75% of all patients with abnormal neurological examinations had secondary type headaches, which accounted for the majority of abnormal neurological examinations. Another risk factor that was identified by 12% of the patients was a family history of headaches, mainly the primary type of headache. Previous works in the literature has reported that, despite potentially life-threatening (LT) secondary headaches being less common (2–15.3%), primitive headaches (21.8–66.3%) and benign secondary headaches (35.4–63.2%) are the most common causes of non-traumatic headache in the emergency department (ED) [23].

This study evaluated the head neuroimaging findings of patients referred to the largest pediatric hospital in Saudi Arabia. In our study, CT scans were abnormal in 50% of the patients, of which 89% were due to a secondary type of headache. This was higher than the findings of a previous study that was conducted in Iran in 2015 by Behzadmehr et al. In the Iranian study, the neuroimaging findings of 217 pediatric patients (aged less than 12 years) revealed that the vast majority of them were normal and only 11.1% of them showed abnormality. Another 136 patients were subjected to an MRI, with 75.7% being normal [17]. In comparison, only 50% had normal CT in our study, while 37.6% of our patients underwent an MRI afterward and only 26% were normal. In 2017, a previous study by Rossi et al. in Italy examined pediatric patients aged below 18 years presenting to the emergency department with a primary headache [3]. A total of 30.0% of the patients reported secondary headaches, 62.1% had primary headaches, and 7.8% had inconsistent diagnoses. Only 1.1% of the patients with secondary headaches had significant problems detected, and they typically had cranial nerve palsy, drowsiness, and strabismus [3].

In 2009, Lateef et al. studied 364 patients between the ages of 2 and 5 years with acute onset of headache in the ED and reported that 84% of them were diagnosed with secondary type of headache, and 16% had primary headache [2]. In addition, one patient with a picture of primary headache had an abnormal CT scan suggestive of brain stem glioma. In our study, nine patients who had a picture of primary type headache (six cases with migraine without aura; two cases with headache not specified; one case with migraine with aura) were incidentally found to have sinusitis in neuroimaging results, which suggests that sinusitis might mimic episodes of primary type headache.

Previous studies from different countries reported that a secondary type headache is the most predominant type of headache [8,9,10]. The rising prevalence of the secondary headache’s underlying cause, which stimulates pain-sensitive areas in the head or neck, may be the reason for the rise in the prevalence of secondary headache [24], or it might be attributed to the fact that there has been an increase in the ED physicians’ ability to differentiate among different types of headache [3]. The majority of secondary headaches are self-limiting, and primarily brought on by upper respiratory tract infections [8,9,10,11,19]. Injuries to the head or neck, vascular disorders of the neck or cranium, non-vascular intracranial disorders, substance abuse or withdrawal, disorders of homeostasis, and disorders of the head, neck, sinuses, teeth, eyes, ears, nose, mouth, or other facial or cervical structures are additional risk factors for secondary headache [24]. One of the possible justifications for the differences in the neuroimaging findings between studies is the type of abnormalities involved; our study included both intra-axial and extra-axial abnormalities, unlike other studies that might involve intra-axial abnormalities only. Another important factor is the age of the study population being investigated.

The primary finding in our study’s CT scans was sinusitis, which was followed by masses, hydrocephalus, cysts, and other conditions such as Chiari malformation, adenoid hypertrophy, and cistern lipoma. Behzadmehr et al. also demonstrated that masses and hematomas were the most common abnormalities on CT scans, followed by cysts and ventriculomegaly [17]. In a systematic review published in 2013, Alexiou et al. looked at the results of 2852 children who had either CT or MRI scans; 14.6% of the individuals showed abnormal findings. Arachnoid cysts, sinusitis, and Chiari I malformation were the most frequently found abnormalities [16]. Cysts were the most frequent abnormal MRI findings stated by Behzadmehr et al., followed by ventriculomegaly, atrophy, and hydrocephalus; here, masses, cysts, hydrocephalus, hematoma, and others including Chiari malformation, glioma, and neurobrucellosis were the most frequently identified abnormalities [17]. After receiving negative CT results, five patients received positive results from subsequent MRI examinations. The MRI revealed a pilocytic astrocytoma that was not visible on the CT scan, while the CT scan revealed a cyst in one patient and hydrocephalus in another with herniation indications. The MRI suggested a new optic nerve glioma in a patient who had previously been diagnosed with NF1 and who now had a headache. The CT scan was unremarkable in this patient. The CT revealed a cyst in other individuals, but an MRI later revealed a hematoma as well. Despite a previous normal CT, the last patient showed a supratentorial lesion. Another prospective cohort study in 2018 by Tsze et al. on healthy children aged in a range of 2 to 17 years found that twenty patients had an abnormal neurological exam which showed that 1% of them had intracranial findings, which support the findings of our research: that there is no major significant relation between abnormal neurological examination and abnormal imaging findings [25].

In comparison to imaging adults, radiological imaging in the pediatric population is a very helpful diagnostic tool, but it also presents a number of unique obstacles. This is due to the following reasons: For lengthier operations, such as MRIs, it is necessary to use sedation or general anesthesia, and specialized training is needed for the acquisition of the images, which needs specialist imaging protocols [26]. The involvement of medical staff, the use of in-depth knowledge and experience in image evaluation and, most importantly, the consideration of radiation dose are all necessary when using ionizing radiation [26]. For the pediatric population, radiation safety and protection are of the utmost importance. Children are nearly ten times more likely than adults to be exposed to cancer-causing factors [27]. Furthermore, since they have a longer lifespan than adults, children are more likely to experience the potential negative consequences of radiation. The As Low As Reasonably Achievable (ALARA) principle should be strictly followed, and CT use should be moderated and used when needed. Depending on the clinical indication, the appropriate imaging modality should be chosen. Except for trauma assessment, MRI is recommended over CT for the majority of the cross-sectional imaging workup in children [26].

One of the most important aspects of ED settings is to control the pain in both subtypes of headaches. Protocols and practices vary widely between hospitals and centers. Medications with appropriate age and weight-based doses should be used to relieve the pain. In our study, the most two commonly prescribed medications in the emergency department were nonsteroidal anti-inflammatory (NSAID) and Acetaminophen, which contributed to 29% and 45%, respectively. Acetaminophen and NSAID drugs are first-line agents upon arrival and discharge from the ED [28]. Ibuprofen has a track record of success and is advised for the treatment of migraine in children and adolescents [29]. Intranasal sumatriptan was found to be inferior to intravenous (IV) ketorolac for treating migraines [30]. Ketorolac was shown to be less effective in treating headaches than prochlorperazine (55.2% vs. 84.8%) in a prospective trial of pediatric migraine patients [31]. A previous study by Raucci et al. reported that only ibuprofen and sumatriptan substantially outperformed placebo at relieving headaches [23].

Symptoms such as nausea and vomiting should also be treated, especially in cases of secondary-type headaches. Dopamine receptor antagonists, such as metoclopramide, and serotonin 5-HT3 receptor antagonists such as Granisetron are generally used in acute settings to control patients’ symptoms. However, they are used as a second line after NSAID and acetaminophen have failed. Other medications can be given in ED settings in secondary type headaches and are based on the underlying causes (e.g., Ceftriaxone given prophylactically in meningitis cases after lumber puncture).

## 5. Conclusions

Neuroimaging testing is a vital diagnostic method for children who suffer from headaches. Children who complained of headaches were more likely to have sinusitis. The likelihood of abnormal CT and MRI findings was higher for the secondary type than the primary type. Those with a positive family history of headaches reported significantly more primary type headaches. CT scans and MRIs are essential when a headache is present along with an abnormal clinical assessment that might indicate an abnormality. Neuroimaging should be considered if there are any clinical anomalies or family histories, and CT use should be moderated and used when needed.

## Figures and Tables

**Figure 1 healthcare-11-00913-f001:**
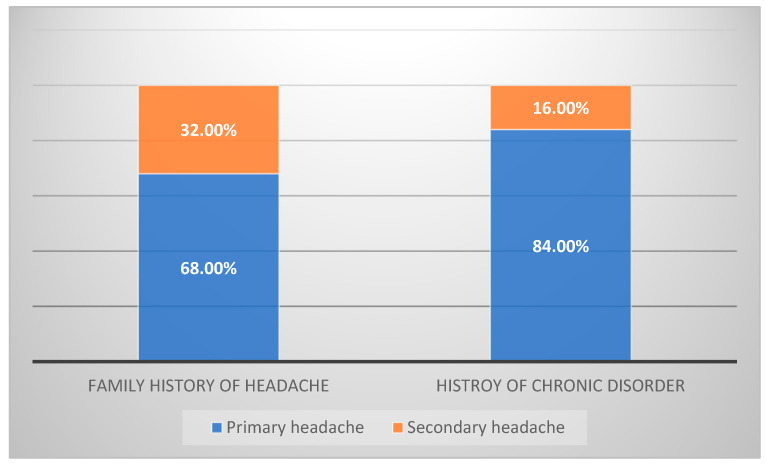
Type of headache with and without history.

**Table 1 healthcare-11-00913-t001:** Basic demographics of children presenting to the ER with headache (n = 263).

Variables	Mean	SD
Age (years)	11.7	3.3
Age of onset (years)	9.3	3.0
Height (cm)	139.1	19.0
Weight (kg)	42.4	19.0
	Frequency	Percentages
Gender		
Males	152	58%
Females	111	42%
Prior diagnosis of headache	25	10%
Family history of headache	31	12%
History of chronic disorder	109	41%
Neurological disorders	55	21%
Respiratory disorders	32	12%
Genetic disorders	18	7%
Malignancy	16	6%
Metabolic disorder	5	2%
Infectious disorders	5	2%
Others	60	23%

**Table 2 healthcare-11-00913-t002:** Physical Examination Findings.

	All Patients ** N = 260		Primary N = 58		Secondary N = 202		*p*-Value *
	Abnormal Examination		Abnormal Examination		Abnormal Examination		
	N	%	N	%	N	%	
Neurological examination	100	39%	25	43%	75	37%	0.43
Mental status changes	14	5%	5	9%	9	5%	0.27
Language	7	3%	1	2%	6	3%	1.0
Cranial nerve	9	4%	3	5%	6	3%	0.43
Motor	24	9%	6	10%	18	9%	0.76
Sensory	2	1%	1	2%	1	1%	0.42
Cerebellar	10	4%	0	0%	10	5%	0.11
Gait	12	5%	1	2%	11	5%	0.29
Reflexes	3	1%	0	0%	3	2%	1.0

* Chi square or Fisher’s exact test was used for the categorical variables. ** The total number of the patients does not add up to 263 due to missed data.

**Table 3 healthcare-11-00913-t003:** Computed tomography and magnetic resonance imaging findings.

CT Findings	All Patients ** N = 260		Headache Type				*p*-Value *
			Primary N = 58		Secondary N = 202		
	N	%	N	%	N	%	
Abnormal CT	131	50%	14	24%	117	58%	<0.001
Mass	18	7%	0	0%	18	9%	0.13
Cyst	14	5%	0	0%	14	7%	0.21
Ventricular dilation	10	4%	0	0%	10	5%	0.61
Hydrocephalus	17	7%	0	0%	17	8%	0.13
Calcification	4	2%	0	0%	4	2%	1.0
VP shunt malfunction	5	2%	0	0%	5	2%	1.0
Infarction	2	1%	0	0%	2	1%	1.0
Edema	6	2%	0	0%	6	3%	1.0
Sinusitis	42	16%	9	16%	33	16%	0.49
Others	85	33%	14	24%	71	35%	0.33
MRI findings							
MRI abnormal	68	26%	2	3%	66	33%	<0.001
Mass A	21	8%	0	0%	21	10%	0.17
Cyst A	12	5%	0	0%	12	6%	0.58
Ventricular/dilation	1	0%	0	0%	1	0%	1.0
Hydrocephalus	8	3%	0	0%	8	4%	1.0
Hematoma	3	1%	0	0%	3	1%	1.0
Sinusitis	2	1%	0	0%	2	1%	1.0
Encephalomalacia	1	0%	0	0%	1	0%	1.0
Hypoplasia of corpus callosum	2	1%	0	0%	2	1%	1.0
Others B	34	13%	2	3%	32	16%	0.21

CT: Computed Tomography; MRI: Magnetic resonance imaging; VP: ventriculoperitoneal; * Chi square or Fisher’s exact test was used for the categorical variables; ** The total number of the patients does not add up to 263 due to missed data.

**Table 4 healthcare-11-00913-t004:** Management stratified by type of headache.

Management Variable	All Patients ** N = 260		Headache Type				*p*-Value *
			Primary N = 58		Secondary N = 202		
Emergency department medications							
	N	%	N	%	N	%	
Anesthetic in ER	25	10%	0	0%	25	12%	0.005
Antiepileptic in ER	8	3%	0	0%	8	4%	0.12
Antiemetic in ER	39	15%	10	17%	29	14%	0.58
NSAID in ER	76	29%	24	41%	52	26%	0.021
Acetaminophen in ER	118	45%	24	41%	94	47%	0.487
Others in ER	15	6%	6	10%	9	4%	0.09
Investigation							
Basic screen tests	218	84%	44	76%	174	86%	0.061
Blood culture	84	32%	11	19%	73	36%	0.014
Urine culture	53	20%	6	10%	47	23%	0.031
LP	32	12%	1	2%	31	15%	0.005
VP shunt X-ray	20	8%	0	0%	20	10%	0.013
Others	49	19%	12	21%	37	18%	0.684
Outpatient medications							
Acetaminophen in OP	83	32%	18	31%	65	32%	0.869
Antiepileptic in OP	8	3%	2	3%	6	3%	0.853
Antibiotics in OP	24	9%	1	2%	23	11%	0.025
Vitamins supplements in OP	16	6%	5	9%	11	5%	0.375
Antihistamine in OP	12	5%	2	3%	10	5%	0.631
Corticosteroids in OP	18	7%	2	3%	16	8%	0.237
NSAIDs in OP	45	17%	20	34%	25	12%	<0.001
Others in OP	21	8%	4	7%	17	8%	0.708
Follow-up							
Admission	135	52%	10	17%	125	62%	<0.001
Follow up	163	63%	41	71%	122	60%	0.153
Persistent Headache	29	11%	12	21%	17	8%	0.009

ER: Emergency Department; OP: Outpatient; LP: lumber puncture: NSAID: Nonsteroidal anti-inflammatory drug; * Chi square or Fisher’s exact test was used for the categorical variables; ** The total number of the patients does not add up to 263 due to missed data.

## Data Availability

Data are available from the corresponding author upon request.

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
