# Peer review of "Diagnostic Yield of Neuroimaging for Headache in a Pediatric Emergency Department: A Single Tertiary Centre Experience"

_healthcare, 2023, doi:10.3390/healthcare11060913_

Round 1

Reviewer 1 Report (Previous Reviewer 5)

Presented work is not in readable format

Abstract needs some technicality not a story

Methodology not upto the mark

A comprehensive analysis must need

Discussion section must re-write no more stories 

The conclusion needs many improvements

Author Response

- We have now tried to address the reviewer comments taking into considerations other reviewers' comments into account. The abstract has now been rephrased (mainly the conclusion). Results checked and modified as appropriate. Methodology further clarified, discussion expanded and conclusion re-written.

Reviewer 2 Report (Previous Reviewer 4)

Although the authors addressed the simple requests in comment 1,5, and 6. The comments below were not sufficiently or clearly addressed.

 2. The author state that multiple studies indicate neuroimaging is unnecessary. What are the reasons behind these studies’ conclusions? If the authors want to support their different conclusion that neuroimaging is necessary, it would be very helpful to provide specific counterargument and evidence against the reasons listed in previous studies.

- Thank you for this comment. We have now addressed this comment in the discussion section and highlighted the risk of neuroimagining on pedatric population and that it should be minimised and that CT use should be moderated and used when needed. Depending on the clinical indication, the appropriate imaging modality should be chosen. Except for trauma assessment, MRI is recommended over CT for the majority of the cross-sectional imaging workup in children, see page 12.

It wasn’t clear to me why previous studies suggest neuroimaging is unnecessary. Is it because of safety concerns and complexity of the imaging procedures?

3. Please reformat Figure 1 into a more presentable form.

- Thank you for this comment, we have now addressed this comment.

I do not see changes in Figure 1.

4. In Table 2 and 3, the number of patients with primary and secondary types of headaches, respectively, don’t add up to the total number of patients, which is 263. Similarly, in Table 3, the number of patients with abnormal MRI scans suffering from primary and secondary types of headaches don’t add up to 69. In Table 2, the neurological examination row has the same issue. Can the authors explain?

- Thank you for this comment. Yes, actually those 7 patients have mixed both primary headache along with secondary headache. As per criteria they meet both. We have now added this point to the results section

This answer doesn’t explain the discrepancy. If 7 patients have both primary and secondary headaches, then in Table 2 and 3, the number of patients with primary and secondary types of headaches, respectively, should add up to 270, which is 7 more than 263. But they only add up to 260. I also do not see a point regarding this comment in the results section.

Author Response

Although the authors addressed the simple requests in comment 1,5, and 6. The comments below were not sufficiently or clearly addressed.

  1. The author state that multiple studies indicate neuroimaging is unnecessary. What are the reasons behind these studies’ conclusions? If the authors want to support their different conclusion that neuroimaging is necessary, it would be very helpful to provide specific counterargument and evidence against the reasons listed in previous studies.

- Thank you for this comment. We have now addressed this comment in the discussion section and highlighted the risk of neuroimagining on pedatric population and that it should be minimised and that CT use should be moderated and used when needed. Depending on the clinical indication, the appropriate imaging modality should be chosen. Except for trauma assessment, MRI is recommended over CT for the majority of the cross-sectional imaging workup in children, see page 12.

It wasn’t clear to me why previous studies suggest neuroimaging is unnecessary. Is it because of safety concerns and complexity of the imaging procedures?

- Yes, due to safety concerns and complexity of the imaging procedures. This is highlighted in the discussion, lines 280-295.

  1. Please reformat Figure 1 into a more presentable form.

- Thank you for this comment, we have now addressed this comment.

I do not see changes in Figure 1.

- Sorry for this unintended mistake, we have now addressed this comment and reformatted the figure number 1, see page 4.

  1. In Table 2 and 3, the number of patients with primary and secondary types of headaches, respectively, don’t add up to the total number of patients, which is 263. Similarly, in Table 3, the number of patients with abnormal MRI scans suffering from primary and secondary types of headaches don’t add up to 69. In Table 2, the neurological examination row has the same issue. Can the authors explain?

- Thank you for this comment. Yes, actually those 7 patients have mixed both primary headache along with secondary headache. As per criteria they meet both. We have now added this point to the results section

This answer doesn’t explain the discrepancy. If 7 patients have both primary and secondary headaches, then in Table 2 and 3, the number of patients with primary and secondary types of headaches, respectively, should add up to 270, which is 7 more than 263. But they only add up to 260. I also do not see a point regarding this comment in the results section.

- Sorry for this unintended mistake, we have now checked all data presented in the manuscript and tables. The total number of patients in table 2, 3, and 4 is 260 due to the availability of the data. The total number does not add up to 263 due to missing data, that is why we highlighted the number of patients in each column as (all patients n= 260, primary n= 58, and secondary n= 202). However, based on the reviewer comment, I have now highlighted this point further in the footnote of the tables and in the results section that we are presenting the data of 260 patients with complete data.

Reviewer 3 Report (Previous Reviewer 2)

I’m happy with the revised version. 

Author Response

Thank you for confirming that you are happy with the current draft.

Round 2

Reviewer 1 Report (Previous Reviewer 5)

Authors failed to present the methodology /model to publish.

No results are presented.

No main contributions are claimed.

No justification why this study needs ?

Author Response

Response to reviewers:

Reviewer 1:

Authors failed to present the methodology /model to publish.

- Thank you for this comment, full details about the study method are available in page 3.

No results are presented.

- Thank you for this comment, results are presented in page 3-7.

No main contributions are claimed.

- Thank you for this comment, the main contribution for this study is highlighted in the study aims in page 2, “to assess neuroimaging findings in pediatric patients coming to the emergency department with headache as a chief complaint. A further purpose of this study was to describe the physical examinations and management procedures for pediatric patients with primary and secondary headache.”

No justification why this study needs ?

- Thank you for this comment, as we highlighted in the introduction section, there are limited local studies in the country that provided neuroimaging findings in children with headache presented to the Emergency department. Examining neuroimaging findings in children with headache is important and could different from country to another due to different environmental factors. Therefore, such type of studies should be conducted and replicated to identify any similarities or difference across patients from different demographic areas and groups.

Reviewer 2 Report (Previous Reviewer 4)

I see improvements compared to the last version. However, the following comments still need to be further addressed.

1. I see a new graph in Fig.1 which looks better, but it only contains partial information compared to the old graph. It has omitted the type of headache without family history and without history of chronic disorder. Also, why the old graph is still included? The title of the figure should be changed as well. “Type of headache with and without history” sounds more accurate.

2. In table 2 and 3, it would be helpful and informative to include data on what abnormalities coexist, given some patients have more than one abnormality.

Author Response

Reviewer 2:

I see improvements compared to the last version. However, the following comments still need to be further addressed.

  1. I see a new graph in Fig.1 which looks better, but it only contains partial information compared to the old graph. It has omitted the type of headache without family history and without history of chronic disorder. Also, why the old graph is still included? The title of the figure should be changed as well. “Type of headache with and without history” sounds more accurate.

- Thank you for this comment, we have now addressed the reviewer comment and changed the title of the figure and removed the old one.

  1. In table 2 and 3, it would be helpful and informative to include data on what abnormalities coexist, given some patients have more than one abnormality.

- Thank you for this comment, we have now addressed the reviewer comment and the most common co-existed abnormalities in table 2 and 3.

This manuscript is a resubmission of an earlier submission. The following is a list of the peer review reports and author responses from that submission.

Round 1

Reviewer 1 Report

radioprotection

This article is well-written and clear.

Abstract: I suggest to include this finding also in the abstract

"There was no statistically significant difference in the proportion of patients diagnosed with primary and secondary headache in term of their neurological examination and headache kinds (p =0.43)."

I suggest to modify the format of the Table 1. The overview of the gender lines is unclear. Maybe you can allign differently the row "gender" or you may include the row "female". Moreover, the subsequent rows (Prior diagnosis of headache, Family history of headache etc.) are referred only to male patients? I suggest to modify the format of table 1.

A neuroimaging rendering or a single exemplifying image could benefit the paper. 

I suggest to include in the discussion a brief paragraph concernign the potential disadvantages of neuroimaging tools, especially in terms of radioprotection for the CT scan in paediatric population. On the other hand, the MRI in ED is not widely common worldwide.

Reviewer 2 Report

overall, a well presented report of significant clinical investigation dealing with paediatric headaches. The report flows well but could benefit from a deeper clinical discussion with more recent literature in childhood headache diagnostic tests and management, such as the paper by Raucci et al in 2019. 

the Stats tests need more explanation, it is not quite clear what tests of difference are carried out in tables and what the significance of these test results are. 

Tables should have legends to describe the results shown.

1. The corresponding author should have * against their name, in the list of authors

2. line 74 CT and MRI are described too late. all abbreviations should be described the first time they are stated.

3. line 97-98, the statement is not clear . "Only 466 of the 8,935 patients who received a CT scan in the ED and reported with a chief complaint of headache had a chief complaint of headache".

4. table 1 and others; how are %ages calculated? are there multi-morbidities? if so, the legend should describe more clearly what they were.

5. table 4 requires a legend to describe the results and show what stats (p values) are showing; what stat test was carried out and what comparisons were made (differences tested). 

6. line 196; maybe best to break the sentence from 'CT' to help with reading.

7. lines 247-256: not sure of this section and what authors are trying to convey. perhaps a rewrite is needed. 

8. line 255; frequently, not frequent

9. line 264; use 2-17 or 2 to 17

10. line 273; describe NSAID when it first appears

11. lines 274-284; unsure what this is trying to convey. 

Reviewer 3 Report

1. The total patient number is 263. There are 58 primary and 202 secondary headaches. This means 7 patients have both. Please clarify. 

2. It will be beneficial to show representative CT/MRI images of the abnormalities since the authors tried to emphasize the neuroimaging finding. It is hard to assess the accuracy of the diagnoses. The standards for these diagnoses should be described. I assume it is based on subjective observation. 

3. Table 3 is confusing. What does CT Normal/Abnormal row mean? 131 should be abnormal. The patient number with Mass through Others is above 300. How many patients have more than one diagnosis? And how does neuroimaging help with that? This is not clear. The same question applies to MRI data.

4. Not all patients received both MRI and CT. If there can be a separate table to summarize both MRI and CT data from the same patient, it will strengthen the significance. 

5. How were the p values calculated? Was there a score from the CT/MRI images? If so, how did they get the scores. In general, the analysis of CT/MRI images was not clear.

Reviewer 4 Report

While the manuscript provides an informative summary of the neuroimaging results of a pediatric population in Saudi Arabia, its data are not sufficient to support one of its key conclusions that neuroimaging is essential for diagnosing pediatric patients with headaches. Besides identifying the cases with life-threatening causes, which are rare, neuroimaging doesn’t seem to provide much additional clinical benefit, such as informing the choice of medication and improving outcomes compared to not performing neuroimaging tests. I suggest the authors either add evidence that demonstrate such clinical benefits or tone down the conclusion that neuroimaging is crucial for pediatric patients with headaches. Furthermore, I have the following recommendations:

1.     In line 81, meningoencephalitides should be meningoencephalitis.

2.     The author state that multiple studies indicate neuroimaging is unnecessary. What are the reasons behind these studies’ conclusions? If the authors want to support their different conclusion that neuroimaging is necessary, it would be very helpful to provide specific counterargument and evidence against the reasons listed in previous studies.

3.     Please reformat Figure 1 into a more presentable form.

4.     In Table 2 and 3, the number of patients with primary and secondary types of headaches, respectively, don’t add up to the total number of patients, which is 263. Similarly, in Table 3, the number of patients with abnormal MRI scans suffering from primary and secondary types of headaches don’t add up to 69. In Table 2, the neurological examination row has the same issue. Can the authors explain?  

5.     In Table 3, “CT Normal/Abnormal” is confusing because the number 131 in the next column is the number of abnormal scans. I suggest changing it to “CT Abnormal”. Similarly, “MRI normal/abnormal” should be “MRI abnormal”.

6.     At the end of line 186, it should be “Table 2” instead of “Table 4” in parentheses, given motor abnormalities and mental status alterations are presented in Table 2.  

Reviewer 5 Report

Authors diagnose Yield of Neuroimaging for Headache in Pediatric  Emergency Department . I have the following major concerns.

No model/methodology presented in section 2 Materials & Methods.

It could be a review paper or analysis etc

Conclusion is not consistent with abstract.

No  justifications provided to support results/findings.